# Classifying the non-metabolic demands of different physical activity types: The Physical Activity Demand (PAD) typology

**Christine Roberts[1], Louise Phillips[2], Clare Cooper[3], Stuart Gray[4], Roy Soiza[5], Julia Allan[3]***

**1** Sport & Exercise Team, University of Aberdeen, Aberdeen, United Kingdom, **2** School of Psychology, University of Aberdeen, Aberdeen, United Kingdom, **3** Institute of Applied Health Sciences, School of Medicine, Medical Sciences & Nutrition, University of Aberdeen, Aberdeen, United Kingdom, **4** Institute of Cardiovascular & Metabolic Health, University of Glasgow, Glasgow, United Kingdom, **5** Institute of Medical Sciences, School of Medicine, Medical Sciences & Nutrition, University of Aberdeen, Aberdeen, United Kingdom

* j.allan@abdn.ac.uk

**Data Availability Statement:** All relevant data are within the paper and its Supporting Information files.

## Abstract

Different physical activity types vary in metabolic demand (intensity), but also in non-metabolic physical demand (balance, co-ordination, speed and flexibility), cognitive demand (attention, memory and decision making), and social demand (social interaction). Activity types with different combinations of demands may have different effects on health outcomes but this cannot be formally tested until such demands can be reliably quantified. The present Delphi expert consensus study aimed to objectively quantify the cognitive, physical and social demands of different core physical activity types and use these scores to create a formal Physical Activity Demand (PAD) typology. International experts (n = 40; experts in cognitive science, psychology, sports science and physiology; 7 different nationalities; 18 male/22 female; M = 13.75 years of disciplinary experience) systematically rated the intrinsic cognitive, physical and social demands of 61 common activity types over 2-rounds of a modified Delphi (expert consensus) study. Consensus (>70% agreement) was reached after 2 rounds on the demands of 59/61 activity types. Cognitive, physical and social demand scores were combined to create an overall non-metabolic demand rating for each activity type, and two-step cluster-analysis was used to identify groups of activities with comparable demand profiles. Three distinct clusters of activities were identified representing activity types with low (n = 12 activities; e.g. domestic cleaning), moderate (n = 23 activities; e.g. tai-chi) and high (n = 24 activities; e.g. football) total non-metabolic demands. These activity types were then organised into a formal typology. This typology can now be used to test hypotheses about if and why physical activity types with different combinations of cognitive, physical and social demands affect health outcomes in different ways.

**Funding:** This work was funded as part of a PhD Studentship awarded to CR by the University of Aberdeen. The funders had no role in study design, data collection and analysis, decision to publish, or preparation of the manuscript.

**Competing interests:** The authors have declared that no competing interests exist.

## Introduction

Physical activity has been described as the best buy in medicine [1]. Evidence from randomized controlled trials shows clear and diverse health related benefits of physical activity, for example, weight loss [2], increased cardiorespiratory fitness [3], better cognitive function [4] and improved quality of life [5]. Even small amounts of physical activity are associated with better health. Across more than 660,000 individuals, people who consistently performed very small amounts of leisure time activity (less than the recommended minimum) had a 20% lower risk of mortality than those who completed none at all [6]. At present however, the beneficial effects of physical activity have been largely studied in relation to metabolic characteristics of activity (frequency, duration and intensity) and little is known about potentially crucial differences in the non-metabolic characteristics of different activity types.

Physical activity types can be differentiated from one another in terms of their underlying cognitive, physical and social demands. For example, walking on flat ground at 2.5 miles per hour is metabolically equivalent (at 3 METs) to dancing a slow waltz [7]. However, these two activities clearly differ in their non-metabolic characteristics. Walking is a simple, routine, relatively automatic activity which can be performed in isolation. Ballroom dancing in contrast is inherently social and requires balance, coordination, flexibility, memory for sequences and real time responses to change (e.g. in tempo, music, partner's actions). Despite these clear differences, the non-metabolic characteristics of different activity types are not commonly considered in health research or are considered only at a very broad level e.g. work versus leisure activity, or aerobic versus strength/resistance activity. While broad classifications may be useful in many aspects of research, they fail to identify the specific non-metabolic cognitive, physical, and social demands of different activity types.

It is an intuitively appealing proposition that activity types with different combinations of underlying demand may have different effects on health outcomes which are related to those demands. For example, activities which require high levels of balance and coordination may be more beneficial in preventing falls in old age. Activities which require high levels of attention and memory may be more beneficial for cognitive health. In one of the few studies to directly address this issue, authors randomized older adults to one of two metabolically equivalent activities—stationary cycling or stationary cycling with a simultaneous cognitive task (cybercycling) and reported that the latter had a greater positive effect on cognitive functioning in older adults [8]. Furthermore, a 2017 systematic review of physical activity interventions and functional independence in older adults found that interventions which used activity types with higher levels of cognitive, social and physical demand had larger beneficial effects on physical performance in old age than activities with lower non-metabolic demands [9]. While the evidence at present is limited, several Cochrane reviews have called for investigation into the differential effects of different types of physical activity on a range of health outcomes, including dementia [10], physical functioning [11] and Health Related Quality of Life (HRQoL; [12]).

If metabolically equivalent activity types with different non-metabolic demands do indeed have differential effects on health outcomes, this has marked implications for both scientific theory and for the design and optimization of interventions involving physical activity. However, in order to systematically understand the effects of the cognitive, social, and physical demands of different types of activity, these non-metabolic demands must be reliably quantified. The present study uses a systematic and structured expert consensus (Delphi) approach plus cluster-analysis to create a formal physical activity demand (PAD) typology which quantifies and then classifies the cognitive, physical, and social demands of common physical activity types.

## Materials and methods

### Design

A modified, two-round, international Delphi consensus study, followed by the generation of a formal Physical Activity Demand (PAD) typology. The study protocol is available at: dx.doi.org/10.17504/protocols.io.yxmvm32kbl3p/v1.

### Ethical approval

The study was reviewed and approved by the University of Aberdeen's College of Life Sciences and Medicine's College Ethics Review Board (CERB/2016/2/1305). All participants provided written informed consent.

### Participants and recruitment

Participants ($N = 40$) were recruited from the UK and Europe via purposive sampling between 1st April 2016 and 30th June 2016, forming two international expert panels: 1. Panel One (n = 20): cognitive psychology experts to provide ratings of the cognitive and social demands of different physical activity types; and 2. Panel Two (n = 20): physiology experts, sport and exercise scientists and physical education specialists to provide data on the non-metabolic, physical demands of different activity types. Suitable participants (identified via professional networks or nominated by other expert participants) were approached directly by email. Inclusion criteria were: graduate qualification and experience of working in a relevant field and good written (English language) communication skills. Across both panels there were 18 males and 22 females, from seven different countries. Of the 40 participants, 37 had advanced degrees (i.e., Master's degree or higher) and a mean 13.75 years of experience in their primary discipline. Participants' time was reimbursed at a rate of £10 per Delphi round. While there is no formal agreement on required panel sizes for Delphi studies, 10 to 18 experts on each panel are recommended [13] plus extra to guard against attrition [14].

### Delphi methodology

The Delphi method is a widely used research methodology which aims to gain consensus of expert opinion in real-world knowledge [15]. Questions about the topic of interest are posed to individual members of an expert panel. Individual responses are then summarized and anonymously recirculated to the panel alongside a reminder of each individual respondent's original data. This allows panel members to: (i) ensure that their views are correctly interpreted; (ii) acknowledge the views of fellow panel members, and (iii) edit their responses (if they wish) after viewing the responses from the rest of the panel. If the panel do not reach adequate agreement after this first round, additional rounds follow until a specified level of agreement is reached. The present study uses a modified Delphi technique [15], whereby preselected items are used to form a structured, round-one questionnaire (as opposed to the open-ended questions used in classic Delphi studies).

### Materials

To create the materials required for the Delphi study, two pieces of development work were undertaken. First, a comprehensive list of common activity types was generated, and second, key non-types of metabolic demand were identified and defined.

**Identifying common activity types.** Ainsworth's Compendium of Physical Activities [7] was used as a starting point to identify all types of physical activity that could be relevant for inclusion in a comprehensive typology of physical activity. The compendium lists 822

individual physical activities alongside their corresponding metabolic (MET) intensity values. Starting with the full 822 items, the compendium was subjected to a six-step systematic process of reduction, which resulted in a final subset of 61 core physical activity types (Fig 1).

In step 1, all activity types that people could not feasibly be randomly allocated to for research purposes (e.g. religious activity, sexual activity, etc) and in step 2, those related to specific occupations (e.g. hairstyling, bookbinding etc) were removed. In step 3 all activities classified as sedentary or very light-intensity (<3.0 METs) were removed on the grounds that very low intensity activities (e.g. breastfeeding, driving a car, etc) fall below the recommended guidelines for physical activity [16] and are unlikely in isolation to contribute measurably to health outcomes. In step 4, duplicate activities, differing only in context or intensity (e.g. bicycling 12–13.9 mph; bicycling on dirt road, bicycling to/from work) were removed, retaining only the version labelled as general or moderate (e.g. bicycling, general) or the version with the median MET value across the set of duplicates. In addition, items that were subcomponents of broader activities were removed. For example, planting trees, shoveling dirt or mud, and planting seedlings or shrubs were deemed subcomponents of the activity gardening and were removed. In step 5, obscure items (e.g., moving an icehouse/setting up drill holes) and those which could contain different combinations of activities and whose content could therefore not be reliably specified (e.g. health club exercise classes) were removed. Finally, in step 6, items relating to activities which were very similar (e.g. ski machine, general; elliptical trainer, moderate effort) were combined into single items for the sake of parsimony (e.g. Cardiovascular machines such as ski machine, elliptical trainer). Irrelevant information was removed from the titles of the 61 remaining items (e.g. 'resistance (strength) training, squats, slow or explosive effort' was changed to 'resistance/strength training'), and clear definitions of each activity type were generated from the literature. The final list of activity types is given in S1 Table.

**Identifying key non-metabolic demands.** One researcher reviewed the existing literature on the conceptual nature of physical activity and compiled an exhaustive list of the possible cognitive, social and (non-metabolic) physical demands involved in different activities. Four researchers then met to discuss and to identify demands from this list that (a) were likely to be present to differing degrees in different types of activity, and (b) that could be clearly defined. Eight demands in total were identified via this process: three cognitive (attention/concentration; memory; decision-making and strategy), four non-metabolic physical (i.e. flexibility; balance; coordination; speeded reactions) and one social (i.e. level of interaction).

The three cognitive demands selected were attention/concentration; memory; and decision-making and strategy. Attention/concentration is maintaining focus on a task, without being distracted by external stimulus or internal thoughts [17, 18]. High levels of attention/concentration are likely to be required during complex tasks or when an individual is presented with information that needs to be processed in real time, for example from music, instructions, environmental conditions, or other people/players. Multiple memory demands can be apparent during physical activity. Procedural memory is an aspect of long-term memory that deals with knowledge of skills required to perform tasks such as riding a bicycle [18]. Working memory is a limited capacity system that involves the ability to hold, manipulate and update information during an immediate memory task [18]. In order to reflect both aspects, memory was defined broadly as recalling complex sequences of actions/movements and/or holding relevant information in mind and updating it as it changes. Activities requiring memory of sequences of motor skills (e.g. dancing), or that require score keeping (e.g. golf) are likely to require high levels of memory. Decision-making and strategy is selecting an appropriate course of action from multiple options; planning actions in advance and anticipating likely outcomes [19]. High levels of decision-making and strategy are likely to emerge when the

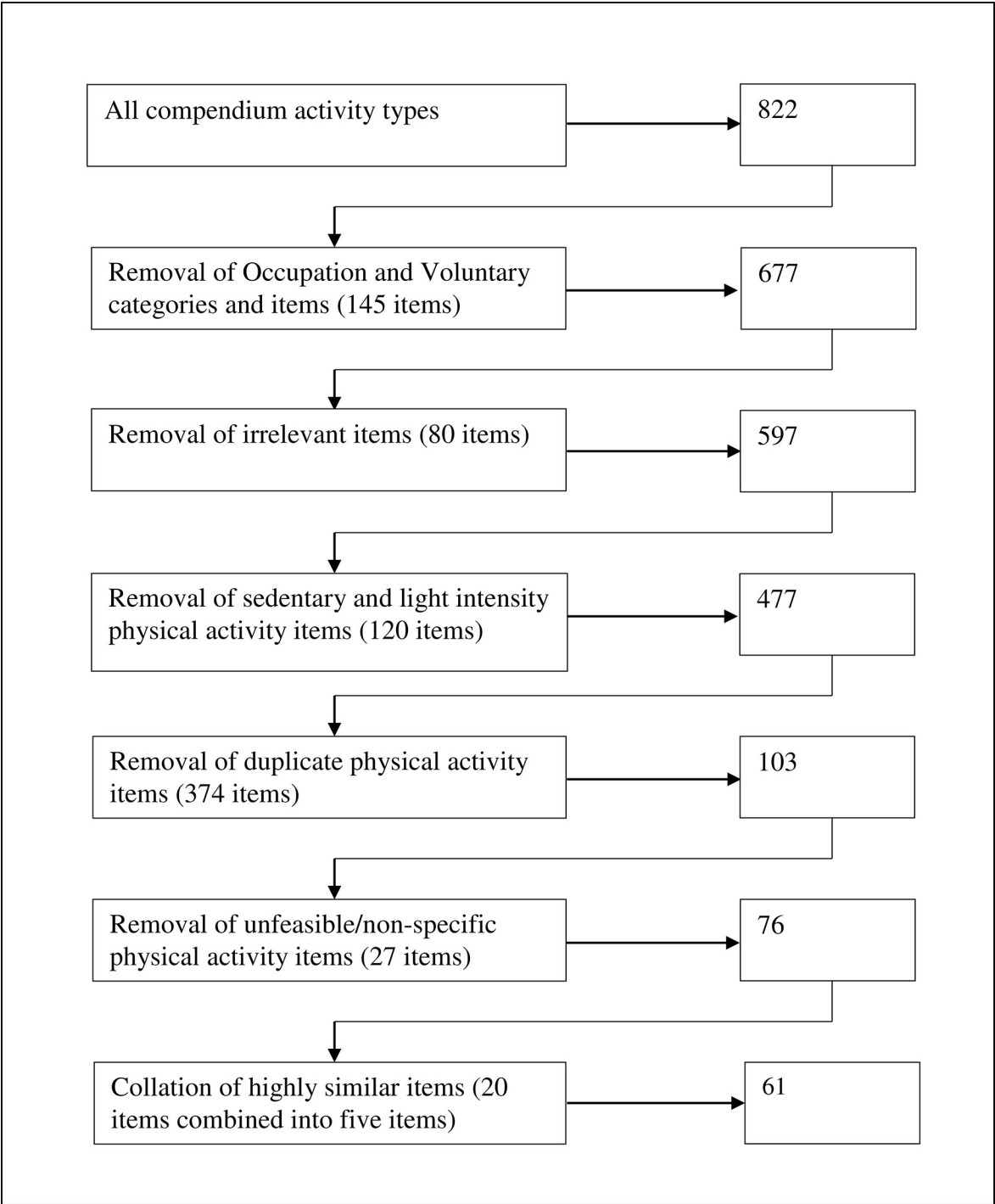

**Fig 1. Flowchart of activity type selection process.**

environment is unpredictable, or in team sports where multiple options are available (e.g. football, basketball).

Four physical but non-metabolic skill-related components of physical activity [20] were selected: flexibility, balance, coordination, and speeded reactions. Flexibility is defined as the

full range of motion available at a joint [21]. Activities requiring high levels of flexibility will include maximal and multiple joint range of motion (e.g. gymnastics). Balance is defined as the ability to maintain equilibrium of the body while stationary or moving [21]. As the stability of a body position reduces as movements become more dynamic, high-level balance activities are likely to include destabilized positioning of the body (e.g. yoga) or environmental conditions that challenge balance ability during dynamic movements (e.g. ice-skating). Coordination is a motor-related ability, defined as a skill that requires use of the senses (e.g. sight and hearing) together with two or more body parts, to perform complex movements and motor tasks smoothly and accurately [21]. While all physical activities require the ability to coordinate limb movements, some activities require only a basic level of coordination (e.g. walking), whereas other activities require more complex movement sequences (e.g. athletics). Speeded reactions are defined as the ability to make a quick response to a fast or unpredictable stimulus [21]. Activities that require greater speeded reactions are likely to be fast-paced activities where the participant is required to quickly respond to a human or environmental stimulus (e.g. table tennis).

A single core social demand, social interaction, was identified, reflecting the necessity of some level of social interaction within each activity. Some activities require more than one person to participate (e.g. tennis), whereas other activities may be performed alone or with others (e.g. running).

## The Delphi questionnaire

The 61 physical activity types and the eight cognitive/physical/social demands identified during the development phase of the study were combined into two modified Delphi questionnaires (one for each expert panel) and pilot tested. The questionnaires were revised in line with pilot feedback (instructions clarified, formatting improved, item order randomized) and finalized. The final questionnaires consisted of a consent form; a section on demographic information; and questions regarding the cognitive/physical/social demands of the 61 core physical activity items (totaling 488 questionnaire items; 244 items per panel). The different physical activity types were listed down the left hand side, and each of the relevant non-metabolic demands (attention/concentration; memory; decision making/strategy and social interaction for panel 1 and flexibility, balance, coordination, and speeded reactions for panel 2) and definitions of these demands were listed in columns along the top. Definitions of each physical activity type were also provided in a separate file for reference. Participants were instructed to rate the extent to which each activity type required each demand (1 = none or little required; 2 = a moderate level required; 3 = a high level required). For the social interaction item, participants were asked to score whether the activity: 1 = is typically performed alone; 2 = can be performed alone or with others; 3 = requires more than one person.

## Procedure

Participants were invited to participate via an email containing a hyperlink to the round-one questionnaire which was hosted online via Snap Survey Software v11.0. Once all panel members had completed the round-one questionnaire, the research team compiled individual reports for each participant displaying their own responses to each item alongside a summary of the panel's average response to each item. Although a cut off of 70% agreement is recommended [22], a more stringent 75% agreement cut off was applied in the first round so that borderline items could be re-reviewed at least once. All items where >75% of the panel gave the same rating were accepted as consensus reached, and all other items were entered into the round-two questionnaire. Round-two questionnaires included (a) a list of the items which

reached agreement in round-one and did not require further rating, (b) an anonymous summary of how the whole panel on average rated each of the remaining items, and (c) a reminder of how the individual participant receiving the questionnaire had rated each of the remaining items. Panelists were instructed to review their own ratings for each item against the panel's average ratings, then choose to either stick with their original rating or (having seen the views of the rest of the panel) amend their rating. Again, once all panel members had completed the questionnaire, scores were summarized. All items where 70% or more of panelists agreed after round-two were accepted as consensus reached. For items that did not reach 70% agreement after two rounds, but where more than 90% of panel members gave adjacent ratings on the 3-point scale (i.e. 1 and 2; or 2 and 3), the average score (i.e. 1.5 or 2.5) was accepted. Remaining items were rejected for inclusion in the final typology on the grounds that expert raters could not reach agreement on their demands.

## Data analysis

After both Delphi rounds were complete, the agreed ratings for the eight cognitive, physical, and social demands of the 61 different physical activities were compiled into a formal typology.

To explore natural groupings of activity types with different profiles of cognitive, physical, and social demands, consensus scores were entered into SPSS version 24.0 and three separate cluster analyses (for cognitive, physical, and social demand) were undertaken. To estimate the total non-metabolic demand of each activity type, the ratings of the eight individual cognitive, physical and social demands were added together to form a total composite or 'multi-demand' score for each of the activity types, ranging from 8–24. These overall demand profile scores were incorporated into the demand typology.

## Results

### Delphi consensus ratings

In round-one of the Delphi, 61 different physical activity types were rated across eight cognitive, physical, and social demands ($N = 488$ ratings). Agreement was reached in 36% ($n = 22$) of attention/concentration ratings; 21% (n = 13) of memory ratings; 28% (n = 17) of decision-making and strategy ratings; 30% (n = 18); 69% (n = 42) of social interaction ratings; 30% (n = 18) of flexibility ratings; 31% (n = 19) of balance ratings; 20% (n = 12) of coordination ratings; and 51% (n = 31) of speeded reaction ratings. Overall, 36% ($n = 174$) of round-one items were agreed by >75% of panel members and were accepted as complete. The remaining 314 non-agreed items were entered into round-two.

In round-two of the Delphi, 280 of the 314 (89%) of the non-agreed scores from round-one were agreed. Agreement was reached on 87% (n = 34) of the remaining attention/concentration ratings; 94% (n = 45) of the remaining memory ratings; 86% (n = 38) of the remaining decision-making and strategy ratings; 100% (n = 19) of the remaining social interaction ratings; 86% (n = 37) of the remaining flexibility ratings; 83% (n = 35) of the remaining balance ratings; 98% (n = 48) of the remaining coordination ratings; and 80% (n = 24) of the remaining speeded reaction ratings. After round-two, 34 items had not reached agreement. However, for 32 of these items, more than 90% of scores were distributed between two adjacent ratings (i.e. 1 and 2; or 2 and 3) indicating that while judges disagreed on the precise extent to which particular demands applied to the activities in question they were in broad agreement across the scale. These items were given a rating representing the mid-point between the two adjacent rating bands (e.g. 50% said that scuba diving required moderate attention/concentration (score = 2), and 50% said it required high attention/concentration (score = 3), so a rating of

moderate-high (score = 2.5) was assigned. Two items did not reach any consistent pattern of agreement; the balance demands of frisbee (low = 40%; moderate = 40%; high = 20%) and the speeded reaction demands of rock climbing (low = 20%; moderate = 25%; high = 55%), therefore, these activities were removed from the typology.

## Cluster analysis

In order to identify activities with different non-metabolic demand profiles, three cluster analyses were undertaken to group activities in terms of their (i) cognitive (i.e. attention/concentration, memory, decision-making and strategy), (ii) physical (i.e. balance, flexibility, coordination, speeded reactions), and (iii) social (i.e. social interaction) demands. This exercise was designed to identify all activity types with specific shared characteristics (e.g. which experts agreed were particularly cognitively, physically or socially demanding). Pre-clustering collinearity assessments via Pearson product movement bivariate correlations were assessed among predictor variables, which indicated no issues regarding collinearity [23]. All analyses had three cluster solutions specified a priori to identify activity types sharing low, moderate, and high patterns of demand within each domain.

Cluster analysis 1 used the three cognitive ratings (attention/concentration, memory, and decision-making and strategy) as predictor variables to determine natural groupings within in the data. Cluster quality was rated as a fair fit ($s = 0.4137$), and the three clusters did not significantly differ in size (low, n = 18 activity types; moderate, n = 25 activity types; high, n = 16 activity types). Activity types in each of the cognitive demand clusters are shown in S2 Table. Those rated as requiring high levels of attention/concentration, memory and decision making/ strategy (n = 16) were; badminton, basketball, football, handball, hockey, hunting, martial arts/ combat sports, polo on horseback, rugby, squash and racquetball, surfing, table tennis, tennis, volleyball, water polo, and windsurfing/sailing.

Cluster analysis 2 used the four physical demand ratings (flexibility, balance, coordination, and speeded reactions) as predictor variables. Cluster quality was rated as a fair fit (s = 0.44). Fewer activity types with low levels of physical demand were identified than moderate or high demand (low, n = 10 activities; moderate, n = 22 activities; high, n = 27 activities). Activity types in each of the three physical demand clusters are shown in S3 Table. Those rated as requiring high levels of flexibility, balance, coordination and speeded reaction (n = 27) were; badminton, basketball, cricket, dancing, diving, figure skating and ice dancing, football, gymnastics, handball, hockey, horseback riding, martial arts/combat sports, polo on horseback, rugby, skateboarding, skating, ice, roller and in-line; skiing, softball and rounders, squash and racquetball, surfing, synchronized swimming, table tennis, tennis, trampolining, volleyball, water polo, and windsurfing/sailing.

Cluster analysis 3 used the single social rating as a predictor variable (social interaction). Cluster quality was rated as a good fit (s = 1.0), but clusters differed markedly in size (low, n = 4 activities; moderate, n = 32 activities; high, n = 23 activities). Activity types in each of the three social demand clusters are shown in S4 Table. Those rated as requiring high levels of social interaction (n = 24) were; aerobics class, badminton, basketball, cricket, croquet, curling, bowls, bowling and shuffleboard, fitness class aqua, fitness class resistance/toning, football, handball, hockey, martial arts/combat sports, playing children's games, polo on horseback, rugby, softball and rounders, spin/RPM/cycle class, squash and racquetball, synchronized swimming, tennis, table tennis, volleyball, and water polo.

### Overall non-metabolic demand

To estimate the total non-metabolic demand of each physical activity type, the mean scores of all eight demands were calculated for each activity type and then rounded to nearest whole number. The distribution of these total scores was then visually inspected and three distinct clusters of activities identified representing activity types with low (*n* = 12 activities), moderate (*n* = 23 activities) and high (*n* = 24 activities) total non-metabolic demands. The activity types in each group are summarized in Table 1 and S5 Table.

### Compilation of the Physical Activity Demand (PAD) typology

The individual and composite non-metabolic demand scores of the 59 core types of physical activity were compiled in the completed PAD typology (Table 1).

## Discussion

The current study systematically quantified the non-metabolic demands of different physical activity types and organized them in a comprehensive typology. This new typology will enable researchers to empirically test for the first time whether activity types with different profiles of non-metabolic cognitive, physical, and social demands have measurably different effects on key health outcomes. The typology also provides a transparent, evidence based and pragmatic method for practitioners to select physical activity types likely to be appropriate for interventions with particular purposes (e.g., to improve balance, or cognitive function).

The cognitive, physical, and social demands of 61 different types of physical activities were coded by two international expert panels. Following two rounds of Delphi, panelists reached agreement on the demands of 59 physical activity types. Cluster analysis was used to identify groups of activity types with low, moderate, or high levels of cognitive, physical, and social demands. While many of the highly demanding activities identified are unlikely to be feasible for inclusion in health interventions (e.g. water polo, windsurfing etc.), several of them (e.g. dancing, bowling, table tennis etc.) are no more metabolically intense than those commonly used in health interventions (e.g. walking) and are still likely to be feasible and accessible enough for use with the general population. While the present study does not test the hypothesis that participation in activity types with different demands will have different effects on health, the typology generated provides a tool that can be used to do so.

All of the activity types rated as having high cognitive and/or physical demand are sports (e.g. football, basketball, table tennis), whereas most of the low cognitive and physical demand activities are not (e.g. cleaning, walking, gardening). Cognitively demanding sports typically require open skills, that is, they are performed in an unpredictable environment where decisions regarding actions must be made in real time. Less cognitively demanding activities typically involve closed skills as they are performed in a predictable fashion [24]. Of the high cognitive demand activities in the present study, all 16 activities are open skilled sports, whereas all 18 low cognitive demand activities are largely closed skilled activities. The present study however moves beyond this simple dichotomy between open and closed skills and considers the more specific and complex demands of each activity type. For example, performing a gymnastics routine is a closed skill, because the routine is fixed in advance and the environment is predictable. However, gymnastics requires concentration, memory, balance, and coordination relative to other comparable activities, resulting in relatively high levels of non-metabolic demand.

Several early studies attempted to classify the demands of different activity types, but most included only broad activity features and did not actually quantify the demands in any way that allowed them to be directly scored or compared. Gentile and colleagues [25] for example

Table 1. Physical Activity Demand (PAD) typology.

| Physical activity | Mental | | | | Physical | | | | | Social | | Total | Multi-demand level |
|---|---|---|---|---|---|---|---|---|---|---|---|---|---|
| | Attention/Concentration | Memory | Decision-making and strategy | Mental Demand | Flexibility | Balance | Coordination | Speeded reactions | Physical Demand | Social interaction | Social Demand | | |
| Aerobics Class | 2 | 1 | 1 | L | 2 | 2 | 2 | 2 | M | 3 | H | 15 | M |
| Archery, non-hunting | 3 | 2 | 1 | M | 1.5 | 1.5 | 2 | 1 | M | 2 | M | 14 | M |
| Army type obstacle course exercise/boot camp training | 2 | 1.5 | 1 | M | 2.5 | 2 | 2 | 2 | M | 2 | M | 15 | M |
| Athletics | 2 | 2 | 1.5 | M | 3 | 2 | 2 | 2 | M | 2 | M | 16.5 | M |
| Badminton | 3 | 2 | 2.5 | H | 3 | 2 | 3 | 3 | H | 3 | H | 21.5 | H |
| Basketball | 3 | 2 | 3 | H | 3 | 2 | 3 | 3 | H | 3 | H | 22 | H |
| Bicycling, not stationary | 2 | 1 | 2 | L | 2 | 2 | 2 | 2 | M | 2 | M | 15 | M |
| Cleaning | 1 | 1 | 1 | L | 1 | 1 | 1 | 1 | L | 1 | L | 8 | L |
| Cooking and food preparation | 2 | 2 | 2 | M | 1 | 1 | 1 | 1 | L | 2 | M | 12 | L |
| Cricket | 2.5 | 2 | 3 | M | 2 | 2 | 3 | 3 | H | 3 | H | 20.5 | H |
| Croquet | 2 | 2 | 2 | M | 1 | 1 | 2 | 1 | L | 3 | H | 14 | M |
| Curling, bowls, bowling and shuffleboard | 2 | 2 | 2 | M | 1 | 2 | 2 | 1 | L | 3 | H | 15 | M |
| CV Exercise Machine | 1 | 1 | 1 | L | 2 | 1.5 | 1 | 1 | M | 1 | L | 9.5 | L |
| Dancing | 2 | 3 | 2 | M | 3 | 3 | 3 | 2.5 | H | 2 | M | 20.5 | H |
| Diving | 3 | 2.5 | 2 | M | 3 | 2.5 | 3 | 2 | H | 2 | M | 20 | H |
| Exergaming e.g. Wii Sports | 2 | 1 | 2 | L | 1.5 | 1.5 | 2 | 2 | M | 2 | M | 14 | M |
| Figure skating and ice dancing | 3 | 3 | 2 | M | 3 | 3 | 3 | 3 | H | 2 | M | 22 | H |
| Fishing | 1 | 1 | 2 | L | 1 | 1 | 1 | 2 | L | 2 | M | 11 | L |
| Fitness class, aqua | 2 | 1 | 1 | L | 2 | 2 | 2 | 1 | M | 3 | H | 14 | M |
| Fitness class, resistance toning | 2 | 1 | 1 | L | 2 | 2 | 2 | 1 | M | 3 | H | 14 | M |
| Football | 3 | 2 | 3 | H | 2 | 2 | 3 | 3 | H | 3 | H | 21 | H |
| Gardening | 1 | 1 | 2 | L | 1 | 1 | 1 | 1 | L | 2 | M | 10 | L |
| Golf | 2 | 2 | 3 | M | 2 | 2 | 3 | 1 | M | 2 | M | 17 | M |
| Gymnastics | 3 | 3 | 1.5 | M | 3 | 3 | 3 | 2.5 | H | 2 | M | 21 | H |
| Handball | 3 | 2 | 3 | H | 2 | 2 | 3 | 3 | H | 3 | H | 21 | H |
| Hockey, field and ice | 3 | 2 | 3 | H | 2 | 3 | 3 | 3 | H | 3 | H | 22 | H |
| Home repair | 2 | 2 | 2 | M | 1 | 1 | 1 | 1 | L | 2 | M | 12 | L |
| Home video/DVD workout | 2 | 1 | 1 | L | 2 | 2 | 2 | 1 | M | 1 | L | 12 | L |
| Horseback riding | 2 | 2 | 2 | M | 2 | 2.5 | 2.5 | 2 | H | 2 | M | 17 | M |
| Hunting | 3 | 2 | 3 | H | 1 | 1 | 2 | 2.5 | L | 2 | M | 16.5 | M |

(Continued)

**Table 1.** (Continued)

| Physical activity | Mental | | | | Physical | | | | | Social | | Total | Multi-demand level |
|---|---|---|---|---|---|---|---|---|---|---|---|---|---|
| | Attention/ Concentration | Memory | Decision-making and strategy | Mental Demand | Flexibility | Balance | Coordination | Speeded reactions | Physical Demand | Social interaction | Social Demand | | |
| Man-powered boating | 2 | 1.5 | 2 | M | 2 | 2 | 2 | 1 | M | 2 | M | 14.5 | M |
| Martial arts/Combat sports | 3 | 3 | 3 | H | 3 | 3 | 3 | 3 | H | 3 | H | 24 | H |
| Orienteering | 2 | 3 | 3 | M | 2 | 1.5 | 2 | 1.5 | M | 2 | M | 17 | M |
| Pilates | 2 | 2 | 1 | M | 3 | 3 | 2 | 1 | M | 2 | M | 16 | M |
| Playing children's games | 2 | 2 | 2 | M | 2 | 2 | 2 | 2 | M | 3 | H | 17 | M |
| Polo, on horseback | 3 | 2 | 2.5 | H | 2 | 3 | 3 | 3 | H | 3 | H | 21 | H |
| Resistance/strength Training | 1 | 1 | 1 | L | 2 | 2 | 2 | 1 | M | 2 | M | 12 | L |
| Rope skipping | 2 | 1 | 1 | L | 2 | 2 | 2 | 2 | M | 2 | M | 14 | M |
| Rugby | 3 | 2 | 3 | H | 2 | 2 | 3 | 3 | H | 3 | H | 21 | H |
| Running, not on treadmill | 2 | 1 | 1 | L | 2 | 2 | 2 | 1 | M | 2 | M | 13 | L |
| Skateboarding | 2.5 | 1 | 2 | L | 2.5 | 3 | 3 | 3 | H | 2 | M | 19 | H |
| Skating, ice, roller and in-line | 2 | 2 | 1 | M | 2 | 3 | 3 | 3 | H | 2 | M | 18 | M |
| Skiing | 3 | 2 | 2 | M | 3 | 3 | 3 | 3 | H | 2 | M | 21 | H |
| Skindiving and scubadiving | 2.5 | 2 | 2 | M | 2 | 1 | 2 | 1 | M | 2 | M | 14.5 | M |
| Softball and rounders | 2.5 | 2 | 2 | M | 2 | 2 | 3 | 3 | H | 3 | H | 19.5 | H |
| Spin/RPM/Cycle class | 1 | 1 | 1 | L | 2 | 1 | 1 | 1 | L | 3 | H | 11 | L |
| Squash and racquetball | 3 | 2 | 3 | H | 2.5 | 2 | 3 | 3 | H | 3 | H | 21.5 | H |
| Surfing | 3 | 2 | 2.5 | H | 2 | 3 | 3 | 3 | H | 2 | M | 20.5 | H |
| Swimming, laps | 1 | 1 | 1 | L | 2 | 1 | 2 | 1 | M | 1 | L | 10 | L |
| Synchronized swimming | 3 | 3 | 2 | M | 3 | 2 | 3 | 1.5 | H | 3 | H | 20.5 | H |
| Table tennis | 3 | 2 | 3 | H | 2 | 2 | 3 | 3 | H | 3 | H | 21 | H |
| Tai Chi | 2.5 | 2 | 1 | M | 2 | 3 | 2 | 1 | M | 2 | M | 15.5 | M |
| Tennis | 3 | 2 | 3 | H | 2 | 2 | 3 | 3 | H | 3 | H | 21 | H |
| Trampolining | 2 | 1 | 1 | L | 3 | 3 | 3 | 2 | H | 2 | M | 17 | M |
| Volleyball | 3 | 2 | 3 | H | 2.5 | 2 | 3 | 3 | H | 3 | H | 21.5 | H |
| Walking, not on treadmill | 1 | 1 | 1 | L | 1 | 1 | 1 | 1 | L | 2 | M | 9 | L |
| Water polo | 3 | 2 | 2.5 | H | 3 | 2 | 3 | 3 | H | 3 | H | 21.5 | H |
| Windsurfing/sailing | 3 | 2 | 3 | H | 2 | 3 | 3 | 3 | H | 2 | M | 21 | H |
| Yoga | 2 | 2 | 1 | M | 3 | 3 | 2 | 1 | M | 2 | M | 16 | M |

*Note.* 1 = none or little required; 2 = a moderate level required; 3 = a high level required; L = low; M = moderate; H = high

created a taxonomy of physical activities that categorized activities along two dimensions: the environmental context (staying the same or changing, i.e. requiring open or closed skills) and the function of actions (whether the body is required to move or remain stationary, and whether the movement requires manipulation of an object). Using this taxonomy, it is possible to identify demanding activities but the resulting classification is heavily biased towards the importance of decision-making and information processing [26]. For example, table tennis is performed in a changing, unpredictable environment (high environmental demand), the body is in a high state of movement (high action demand) and the player is required to maneuver a bat in order to hit the ball (high object manipulation demand). Gymnastics, however, is performed in an unchanging, predictable environment (low environment demand), where the body is required to perform complex movement sequences (high action demand), often using apparatus (object manipulation). Gentile et al.'s taxonomy would consequently rate table tennis as the more demanding of the two activities. The present typology allows for a more nuanced distinction to be made between the different activities as each different cognitive, physical, and social demand (e.g., balance, coordination, memory etc.) is considered separately. The present classification also gives each demand a numerical rating, enabling direct comparisons to be made between activity types on multiple different dimensions.

In addition to the core cognitive and physical demands, the present typology also includes the social interaction inherent in each activity. This is important as it is well established that social interaction is associated with a number of important health outcomes, including self-rated health [27], longer survival [28] and quality of life in old age [29] and may be one of the mechanisms thorough which physical activity positively impacts health.

Twenty-four physical activities were identified as having high overall non-metabolic demands (e.g. badminton, basketball, cricket, dancing, diving). Such activities require participants to practice demanding cognitive and physical skills in social environments. Preliminary evidence suggests that such activities have larger beneficial effects on physical performance in old age [9] and the present typology will allow researchers to empirically investigate whether similar patterns are seen in other diverse health outcomes such as quality of life, physical health and cognitive function.

The present study is the first to systematically quantify the non-metabolic demands of a comprehensive set of types of physical activity. As such, it provides researchers with the tool required to address calls from the literature to investigate the effects of physical activity type on a range of key health outcomes [10–12]. The methods employed were systematic and every effort was made to include all potentially relevant activity types to maximize the coverage, relevance, and utility of the final typology. In terms of limitations, the Delphi process and the selection of the eight non-metabolic demands used in the present study were necessarily subjective. However, an appropriately sized sample was recruited, and the included demands were selected from an exhaustive literature review to minimize any issues arising from these limitations.

## Conclusions

The present study systematically generated a comprehensive typology of the cognitive, physical, and social demands of 59 different types of physical activity. Scores are provided for specific non-metabolic demands (e.g. attention, memory, balance, etc.) for summary cognitive, physical, and social demands, and also for total, overall non-metabolic demands. It is hoped that this will maximize the range of possible applications. Further research can now use this typology to establish whether different types of physical activity have differential effects on key health outcomes.

## Supporting information

**S1 Table. Core 61 physical activity items.**
(DOCX)

**S2 Table. Results of cluster analysis 1 (mental demands of physical activity).**
(DOCX)

**S3 Table. Results of cluster analysis 2 (physical demands of physical activity).**
(DOCX)

**S4 Table. Results of cluster analysis 3 (social demands of physical activity).**
(DOCX)

**S5 Table. Physical activities categorized as low, moderate, or high multi-demand level.**
(DOCX)

**S1 Data.**
(XLSX)

**S2 Data.**
(XLSX)

## Acknowledgments

We thank the members of the expert panel for giving up their time.

## Author Contributions

**Conceptualization:** Christine Roberts, Louise Phillips, Julia Allan.

**Data curation:** Christine Roberts.

**Formal analysis:** Christine Roberts.

**Funding acquisition:** Christine Roberts, Louise Phillips, Clare Cooper, Stuart Gray, Roy Soiza, Julia Allan.

**Investigation:** Christine Roberts, Julia Allan.

**Methodology:** Christine Roberts, Louise Phillips, Julia Allan.

**Project administration:** Christine Roberts.

**Supervision:** Louise Phillips, Clare Cooper, Stuart Gray, Roy Soiza, Julia Allan.

**Writing – original draft:** Christine Roberts, Julia Allan.

**Writing – review & editing:** Louise Phillips, Clare Cooper, Stuart Gray, Roy Soiza, Julia Allan.

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
