## [Decision Letter · Decision Letter 0]

2 Aug 2023

PONE-D-23-21859Classifying the non-metabolic demands of different physical activity types: the Physical Activity Demand TypologyPLOS ONE

Dear Dr. Allan,

Thank you for submitting your manuscript to PLOS ONE. After careful consideration, we feel that it has merit but does not fully meet PLOS ONE’s publication criteria as it currently stands. Therefore, we invite you to submit a revised version of the manuscript that addresses the points raised during the review process.

We look forward to receiving your revised manuscript.

Kind regards,

Jovan Gardasevic

Academic Editor

PLOS ONE

 Journal Requirements:

   "This work was funded as part of a PhD Studentship awarded to CR by the University of Aberdeen"

4. We are unable to open your Supporting Information file Data2_Typology_Cluster.sav. Please kindly revise as necessary and re-upload.

Additional Editor Comments:

Dear Author,

Please revise the manuscript based on the reviewers' comments.

Reviewers' comments:

Reviewer's Responses to Questions

**Comments to the Author**

1. Is the manuscript technically sound, and do the data support the conclusions?

Reviewer #1: Yes

Reviewer #2: Yes

2. Has the statistical analysis been performed appropriately and rigorously? 

Reviewer #1: Yes

Reviewer #2: Yes

3. Have the authors made all data underlying the findings in their manuscript fully available?

Reviewer #1: Yes

Reviewer #2: Yes

4. Is the manuscript presented in an intelligible fashion and written in standard English?

Reviewer #1: Yes

Reviewer #2: Yes

5. Review Comments to the Author

Reviewer #1: I have carefully read the manuscript and my opinion is that the manuscript has a merit to be published in your reputable journal with some minor corrections. The manuscript is original, informative and readable. This study uses a systematic and structured expert consensus (Delphi) approach plus cluster-analysis to create a formal physical activity demand (PAD) typology which quantifies and then classifies the cognitive, physical, and social demands of common physical activity types. The introduction is well written, while materials and methods section is very well prepared and organized according to contemporary methodological rules. At the end, I have no amendments on results and discussion as well as conclusion part but I would recommend to the authors to exclude ethical approval subsection from the method section as it is technical information that should be placed in some additional section such as acknowledgement or something similar. Lastly, I would recommend you to accept this manuscript right after I confirm the authors revise it in the adequate manner.

Reviewer #2: The abstract does not have a proper structure. It is written as guidelines. Nothing about the sample..., the method of measurement...

Interesting study. The authors show an exceptional level of experience. I recommend that the study be published in its current form. Of course with Abstract corrections

6. PLOS authors have the option to publish the peer review history of their article (what does this mean?). If published, this will include your full peer review and any attached files.

Reviewer #1: **Yes: **Stevo Popovic

Reviewer #2: No

---

## [Author Response · Author response to Decision Letter 0]

4 Aug 2023

Please find attached our full response in the document 'Response to Reviewers'

---

## [Decision Letter · Decision Letter 1]

5 Sep 2023

Classifying the non-metabolic demands of different physical activity types: the Physical Activity Demand (PAD) typology

PONE-D-23-21859R1

Dear Dr. Allan,

We’re pleased to inform you that your manuscript has been judged scientifically suitable for publication and will be formally accepted for publication once it meets all outstanding technical requirements.

Kind regards,

Jovan Gardasevic

Academic Editor

PLOS ONE

Additional Editor Comments (optional):

Dear Author,

after a resubmission of the revised manuscript based on the comments of the reviewers and based on a new review by the same reviewers, as well as on the opinion of the Academic Editor, the manuscript is now ready for publication in this respectable journal.

Reviewers' comments:

Reviewer's Responses to Questions

**Comments to the Author**

1. If the authors have adequately addressed your comments raised in a previous round of review and you feel that this manuscript is now acceptable for publication, you may indicate that here to bypass the “Comments to the Author” section, enter your conflict of interest statement in the “Confidential to Editor” section, and submit your "Accept" recommendation.

Reviewer #1: All comments have been addressed

Reviewer #2: All comments have been addressed

2. Is the manuscript technically sound, and do the data support the conclusions?

Reviewer #1: Yes

Reviewer #2: Yes

3. Has the statistical analysis been performed appropriately and rigorously? 

Reviewer #1: Yes

Reviewer #2: Yes

4. Have the authors made all data underlying the findings in their manuscript fully available?

Reviewer #1: Yes

Reviewer #2: Yes

5. Is the manuscript presented in an intelligible fashion and written in standard English?

Reviewer #1: Yes

Reviewer #2: Yes

6. Review Comments to the Author

Reviewer #1: I have carefully read the revised manuscript and my opinion is that the manuscript has a merit to be published in your reputable journal in current format. The manuscript is original, informative and readable. This study uses a systematic and structured expert consensus (Delphi) approach plus cluster-analysis to create a formal physical activity demand (PAD) typology which quantifies and then classifies the cognitive, physical, and social demands of common physical activity types. The introduction is well written, while materials and methods section is very well prepared and organized according to contemporary methodological rules. At the end, I have no amendments on results and discussion as well as conclusion part but I recommended to the authors to exclude ethical approval subsection from the method section as it is technical information that should be placed in some additional section such as acknowledgement or something similar and they did. Lastly, I would recommend you to accept this manuscript.

Reviewer #2: (No Response)

7. PLOS authors have the option to publish the peer review history of their article (what does this mean?). If published, this will include your full peer review and any attached files.

Reviewer #1: **Yes: **Stevo Popovic

Reviewer #2: No

---

## [Editor Report · Acceptance letter]

12 Oct 2023

PONE-D-23-21859R1 

Classifying the non-metabolic demands of different physical activity types: the Physical Activity Demand (PAD) typology 

Dear Dr. Allan:

I'm pleased to inform you that your manuscript has been deemed suitable for publication in PLOS ONE. Congratulations! Your manuscript is now with our production department. 

Kind regards, 

on behalf of

Dr. Jovan Gardasevic 

Academic Editor

PLOS ONE